# Magnetic Hyperthermia Therapy for High-Grade Glioma: A State-of-the-Art Review

**DOI:** 10.3390/ph17030300

**Published:** 2024-02-26

**Authors:** Benjamin Rodriguez, Daniel Rivera, Jack Y. Zhang, Cole Brown, Tirone Young, Tyree Williams, Sakibul Huq, Milena Mattioli, Alexandros Bouras, Constantinos G. Hadjpanayis

**Affiliations:** 1Icahn School of Medicine at Mount Sinai, New York, NY 10029, USA; daniel.rivera@icahn.mssm.edu (D.R.); jack.zhang@icahn.mssm.edu (J.Y.Z.); cole.brown@icahn.mssm.edu (C.B.); tirone.young@icahn.mssm.edu (T.Y.); 2Sinai BioDesign, Department of Neurosurgery, Mount Sinai, New York, NY 10029, USA; willit13@rpi.edu; 3Rensselaer Polytechnic Institute, Troy, NY 12180, USA; 4Department of Neurological Surgery, UPMC, Pittsburgh, PA 15213, USA; 5Brain Tumor Nanotechnology Laboratory, UPMC Hillman Cancer Center, Pittsburgh, PA 15213, USA; mattiolim3@upmc.edu (M.M.); bourasa@upmc.edu (A.B.); 6Center for Image-Guided Neurosurgery, Department of Neurological Surgery, University of Pittsburgh School of Medicine, Pittsburgh, PA 15213, USA

**Keywords:** MHT: Magnetic Hyperthermia Therapy, glioma, HGG = High Grade Glioma, hyperthermia, MNP = Magnetic Nanoparticle

## Abstract

Magnetic hyperthermia therapy (MHT) is a re-emerging treatment modality for brain tumors where magnetic nanoparticles (MNPs) are locally delivered to the brain and then activated with an external alternating magnetic field (AMF) to generate localized heat at a site of interest. Due to the recent advancements in technology and theory surrounding the intervention, clinical and pre-clinical trials have demonstrated that MHT may enhance the effectiveness of chemotherapy and radiation therapy (RT) for the treatment of brain tumors. The future clinical success of MHT relies heavily on designing MNPs optimized for both heating and imaging, developing reliable methods for the local delivery of MNPs, and designing AMF systems with integrated magnetic particle imaging (MPI) for use in humans. However, despite the progression of technological development, the clinical progress of MHT has been underwhelming. This review aims to summarize the current state-of-the-art of MHT and offers insight into the current barriers and potential solutions for moving MHT forward.

## 1. Introduction

High-grade glioma (HGG), the most prevalent primary brain malignancy, presents a formidable challenge due to its resistance to standard treatment regimens, namely, maximal safe resection supplemented by adjuvant temozolomide (TMZ) chemotherapy and fractionated radiotherapy (RT) [1,2]. The infiltrative nature of HGG prevents complete resection, and the presence of resistant tumor cells within the surrounding healthy brain is a major driver of recurrence [3,4]. In over 80% of cases, HGG recurs locally, typically within two centimeters of the resection cavity [4]. This local recurrence underscores the potential of therapies that target the affected area directly. Given the dire statistics—a median survival of 15 months and below a 5% five-year survival rate [2,5]—more effective treatment approaches are desperately needed.

Magnetic hyperthermia therapy (MHT) is a re-emerging treatment approach for HGGs consisting of local heat generation in the tumor region through the direct delivery of magnetic nanoparticles (MNPs), which are activated by exposure to an external alternating magnetic field (AMF) [6,7,8,9,10]. The major advantages of MHT are its ability to focus heating on a small volume of a tumor without damaging surrounding tissue and its lack of ionizing radiation. Magnetic hyperthermia therapy leverages the unique cellular responses to heat, as the 42–45 °C hyperthermia it induces selectively causes tumor cell death while preserving healthy cells. This effect is facilitated by the hyperthermic environment in the tumor, which activates heat shock proteins and stimulates a strong immune cell response, enhancing the antitumor effect [11,12].

Although MHT is not yet a standard treatment for HGGs, its potential integration into existing treatment protocols is promising, especially as an adjunct therapy. This review explores key aspects of MHT imaging and treatment through the lens of a proposed treatment workflow [13]. Additionally, we address the challenges in transitioning MHT to clinical use and the ongoing efforts to resolve these issues. Our goal is to provide a comprehensive overview of the latest advancements in MHT, aiming to stimulate the further development and integration of this emerging technology into clinical practice for the safe and effective treatment of HGGs.

## 2. MHT Workflow

The workflow used to guide this review was proposed by Healy et al. [13]. This workflow involves both a clinician and a medical physicist working in concert to (1) obtain imaging (CT, MRI) to direct initial treatment planning (tumor margin, anatomy for NP delivery), (2) select optimal nanoparticles for treatment, (3) deliver chosen nanoparticles, (4) obtain post-procedural imaging (MPI) to ensure proper implantation, and (5) implement MHT using AMF (Figure 1). This review will focus on steps seeing the most innovation: nanoparticle selection and delivery, MPI imaging, and AMF treatment.

### 2.1. Nanoparticles

MHT efficacy relies heavily upon designing MNP constructs capable of generating a significant thermal dose at non-toxic concentrations. The thermal dose is dependent on the temperature achieved within the lesion of interest and the time for which that temperature is sustained [14]. Specifically, the thermal dose is calculated based on the maximum temperature achieved across 90% of the lesion (T90). Antitumor effects have been well described when the T90 is between 40 and 45 °C (mild hyperthermia) and >50 °C (thermal ablation) [15]. MHT for HGG aims to heat tissues within the lower range, between 42 and 45 °C [16]. 

MNPs’ heating capacity derives from their magnetic properties, which are influenced by the composition, size, density, and shape [17] an overview of MNP properties is shown in Table 1. Specifically, MNPs are superparamagnetic, meaning that the torque of the AMF acts upon the coupled electron spins of the ferromagnetic nanoparticles. This leads to MNPs having a much higher susceptibility and therefore a higher heating efficiency [18]. MNP heating efficiency is often described in specific loss power (SLP), defined as the measured thermal loss normalized to the mass or volume of magnetic material [19,20]. However, SLP can be context-dependent and therefore inconsistently measured at different frequencies [21]. Although, according to Rosenweig’s model, at low frequencies (10^5^–10^6^ Hz) such as those used in MHT, the out-of-phase component of susceptibility can be held constant [22], allowing for the utilization of the parameter intrinsic loss power (ILP). ILP provides a system-independent measurement parameter to measure MNP heating efficiency. To date, the most common materials used for MNP synthesis include pure metals (e.g., iron, cobalt, nickel), alloys (e.g., FeCo, alnico), and oxides (e.g., Fe_3_O_4_, g-Fe_2_O_3_, CoFe_2_O_4_). Magnetic iron oxide nanoparticles (MIONPs) are used most often in treating glioma due to their superior biocompatibility, established heating profile, and relatively low production cost [23]. With regard to MNP design for MHT specifically, substantial research has been dedicated to optimizing the heating efficiency in terms of MNP size, with the most common and effective MNPs reported to have a diameter between 10 and 20 nm [24,25,26,27]. Other groups have studied the efficacy of various shapes and arrangements of MNPs, showing that both cubic [27] and chain-like arrangements of MNPs [28] heated more effectively compared to spherical and randomly arranged MNPs, respectively. An additional finding was that an MNP suspension solution’s viscosity also impacts heating, with a higher viscosity resulting in decreased SLP [28].

Another important consideration for MNP optimization is ensuring they do not aggregate and precipitate once introduced. To achieve this, MNPs are often conjugated with various compounds and polymers, such as polyethylene glycol (PEG) and chitosan, to increase their chemical stability and solubility [29,30,31]. Numerous other surface coatings have also been applied for similar reasons [32,33,34,35,36]. Carbon-coated MNPs, in particular, have attracted significant attention due to their improved thermal stability compared to uncoated counterparts [37]. Silica is also frequently used, due to its biocompatibility and ability to limit MNP aggregation [38,39,40]. However, it has been shown that interactions between silica and the MNP surface may result in a reduction in saturation magnetization by 32%, thus decreasing heating efficiency [41]. This effect may be variable, as others have recently demonstrated that silica coating may reduce saturation magnetization by as low as 8% and may even result in an increase of as much as 14% [42]. Similarly, organic compounds such as PEG, which are used to reduce non-specific interactions between MNPs and proteins, may affect MNP geometry and therefore their magnetic properties [43,44]. These two examples highlight the complexity of MNP design and the trade-offs between biological stability and magnetic properties that must occur to synthesize stable, non-toxic, and effective MNPs.

Lastly, it is important to address the growing biomedical application of multi-core MNP systems. Thus far, our discussion has focused on single-core systems in which each nanoparticle is composed of a single magnetic core. In contrast, multi-core systems fix multiple magnetic cores together within a single matrix, allowing for additional intra- and inter-particle magnetic dipolar interactions [45]. Numerous studies have found that these multi-core systems exhibit superior heating for magnetic hyperthermia [46,47,48]. Unsurprisingly, the process for synthesizing these multi-core MNPs is complex and depends on the precise control of numerous parameters ranging from temperature and stirring conditions to reagent and surfactant concentrations [49]. As a result, batch-to-batch reproducibility and the up-scaling of production is a major challenge. A significant amount of research is devoted to refining multi-core MNP synthesis methods [49].

### 2.2. MPI

Beyond their ability to serve as therapeutic heating agents, MNPs can serve as diagnostic agents as well. MNPs have been used as MRI contrast agents due to their exceptionally high relaxivity [58,59]. However, at the high MNP concentrations required for MHT, MRI is not possible due to signal saturation, which results in a “black hole” susceptibility artifact on MRI that obscures all relevant anatomy [13]. This problem could be addressed by magnetic particle imaging (MPI), an emerging tomographic technique that may enable the real-time 3D imaging of MNPs at high therapeutic concentrations [60,61,62]. Briefly, MPI systems operate by generating strong magnetic field gradients that contain a specific area of low field strength, known as the field-free region (FFR). Rapidly passing the FFR over MNPs causes their magnetization to flip, creating a detectable signal. Importantly, biological tissue does not produce a significant signal in response to the low-amplitude magnetic fields used in MPI, giving the MNPs ideal contrast independent of their depth within the tissue [63]. However, this also means that MPI is unable to visualize underlying tissue anatomy, necessitating anatomic coregistration with a CT scan or MRI. The unique benefits of MPI include improved imaging signal-to-noise ratios, high spatial and temporal resolutions, the linear quantification of the number of MIONPs regardless of the tissue depth, and the ability to image MNPs at concentrations typically used for MHT (50–100 mg of Fe per g of tissue) [62].

With regard to MHT, preliminary studies have used MPI for image-guided MHT in vivo [64] and have even designed dual MPI–MHT systems [65]. One current limitation of MHT is that it is not possible to focus the high-frequency fields needed for MHT (>300 kHz) to specific regions of the body, thus posing the risk of the off-target heating of MIONPs that may have unintentionally migrated elsewhere in the body, such as the liver. However, one group demonstrated that by employing an MPI gradient field, they were able to achieve highly localized heating to a region just a few millimeters in size, preventing the heating of MNPs outside of the FFR [64,65]. The underlying physics of MPI can further be applied to enable real-time, noninvasive magnetic nanothermometry (MNT) during MHT. To understand this first requires understanding that the magnetization vector of MNPs changes when they are exposed to an AMF and upon the subsequent removal of that AMF signal. The timing of these changes in the MNP magnetization state is influenced by the temperature of the sample. Therefore, it is thought that by comparing the timing differences of changes in MNP magnetization against the time scale of the MPI device, the temperature of the sample can be estimated [13]. In 2023, one group designed a prototype of such a system, with promising preliminary data showing the ability for combined MHT–MNT–MPI in situ [66]. This is a significant advancement, as previous clinical trials studying MHT in the brain have relied on the insertion of invasive intracranial thermal probes for thermometry, effectively negating a key advantage of MHT—its non-invasiveness.

To date, MPI has shown great promise in clinical applications ranging from angiography to cancer theranostics and molecular imaging [67,68]. Recently, there has been a significant effort to develop MNPs optimized for MPI [69,70]. Superparamagnetic iron oxide nanoparticles (SPIONs) are the most effective agents for MPI, since their superparamagnetism enables high-order harmonics of excitation frequencies required for MPI [71]. The current challenge remains customizing MNPs with properties that enable both effective hyperthermia and MPI in vivo. Additionally, the upscaling of MPI systems for human use is needed, although this is being actively pursued [67,71].

### 2.3. Nanoparticle Delivery

The blood–brain barrier (BBB) is an important consideration for intracranial MHT, as it limits the efficacy of systemic MNP delivery [72]. Intracranial lesions must therefore be accessed directly through local delivery [69] by way of: direct intracavitary implantation following surgical resection, convection-enhanced delivery (CED) using a stereotactically placed catheter, or direct stereotactic injection (Figure 2). The underlying principle is a balance between maximizing intratumoral MNP delivery while minimizing reflux and undesired off-site toxicities [69,70]. Reflux is the retrograde flow of fluid back up the catheter or cannula’ this can lead to MNPs localizing in healthy tissue and causing damage when heated [73]. A further consideration in the delivery of MNPs is to preferentially choose minimally invasive techniques or those that can already fit within the workflow of another vital intervention.

Direct implantation is a modality that completely avoids the risk of reflux and allows for direct visualization as the resection cavity is still open and MNPs are pasted directly within [74]. Once the resection is complete, the surgeon will “paste” a viscous solution of MNPs on the wall of the resection cavity either by directly applying the solution [74] or by utilizing hydroxycellulose mesh and fibrin glue to layer the MNPs and provide enhanced stability [75]. Residual tumor cells then take up the MNPs, most often via the clathrin- and caveolae-mediated endocytosis pathways [74,76,77]. This modality has already been studied in two human trials [74,75]. This can be an invasive approach; however, if it fits into the workflow of the resection, then there is no excess trauma being done to the patient. With all of the risks and benefits considered, direct implantation is an ideal adjuvant modality for treating primary HGGs, where maximum possible resection is the primary standard-of-care treatment [78]. However, given recent data suggesting that reoperation does not necessarily provide an improved EOR [79], resection may not be the most effective intervention for patients with rHGG [78]. This eliminates the opportunity to deliver nanoparticles via direct implantation, and a different technique is needed for these patients.

Direct stereotactic injection is the most minimally invasive choice; only a small cannula is stereotactically inserted into the lesion, often without a preceding debulking procedure [80]. In the two human studies performed using this technique [50,80], the cannula was placed stereotactically multiple times within the lesion, dispensing small volumes of MNPs throughout the lesion. The authors delivered 0.4–1.4 mL of NPs over a 30–40 s interval 8–10 mm apart; the slow delivery and small volumes are tactics for minimizing fluid flow to prevent reflux along the cannula [80]. Stereotactic injection provides an option for rHGG in that it does not require a preceding resection and it is minimally invasive. However, the tactic required to minimize reflux (small volume and slow infusions) is a limiting factor in the efficacy of this technique. Direct injection is a viable option, but the ideal modality would allow for higher-volume infusions at an increased rate.

CED infusion is currently the most effective technique for the infusion of MNPs in patients lacking a preceding resection. The modality utilizes a burr hole with the stereotactic insertion of one or more catheters into the target lesion. An infusion pump then generates a pressure gradient at the catheter tip, infusing MNPs directly into the brain [81]. This form of infusion relies on bulk flow rather than diffusion to displace extracellular fluid and prevent reflux. CED mitigates the need for small-volume, slow infusions that are common with stereotactic injection [82]. CED is the most common delivery technique to date; robust literature exists describing CED in both large animal models and humans [83]. In three reported canine studies [70,84,85] and three human studies [74,86,87], MNPs were infused via a stereotactically placed catheter. However, despite CED being an improvement over direct injection, there is still a need to improve delivery techniques to allow for faster delivery and larger infusions, and further preclinical research is required to optimize nanoparticle delivery and minimize reflux.

### 2.4. AMF

Following safe and accurate MNP delivery, exposure to an alternating magnetic field (AMF) is required to excite the MNPs and generate hyperthermia. AMF is a magnetic field (MF) with an amplitude that varies over time [88] and generates MNP heating primarily through hysteresis loss. Briefly, in this process, AMF exposure induces cycles of magnetization and demagnetization, as the magnetic domains present within the MNP continuously switch to reorient with the fluctuating magnetic field [89]. Thermal energy production then primarily occurs during the return to equilibrium (relaxation) of individual magnetic domains, which, in order to overcome a rotational energy barrier, experience energy loss in the form of heat [22,89]. In Néel relaxation, energy loss occurs as the magnetic vectors reorient against the atomic lattice of the magnetic core [89,90]. In Brownian relaxation, energy loss occurs as entire MNPs themselves experience friction from rotating within the surrounding medium [89,90].

AMF parameters are an essential consideration in MHT, as therapeutic heating only begins upon the exposure of MNPs to the AMF. In accordance with the model of AMF-induced MNP heat generation described by Rosensweig et al., the quantity of electromagnetic energy that can be converted into heat increases with the MF amplitude and frequency [22,91]. In theory, target lesion destruction is achieved by maximizing these two parameters. In clinical practice, however, safety considerations impose an upper limit on these parameters. AMF can generate electric current loops (known as eddy currents) which can cause the thermal injury of healthy tissue and negatively impact MHT efficacy [92,93]. In prior clinical studies, MHT tolerance has been shown to be limited by headaches, which could potentially be mediated by temporary, heat-related increases in intracranial pressure [50,93,94].

To date, no long-term adverse effects of AMF or MHT treatment have been reported, and much of the existing data suggest that MHT is an overall safe intervention [9,10,14,95]. Most AMF generators approved for use in MHT produce MFs with frequencies far below the dangerous radio frequency range [10,96] and orders of magnitude below the frequencies used in routine MRI [97]. Even so, no universal guidelines detailing the upper limits of safe AMF exposure currently exist. Atkinson and Brezovich were among the first to investigate safe frequencies in MHT [98,99]. Their clinical studies on healthy patients culminated in the Atkinson–Brezovich criterion, which established a maximum MF-frequency product (H × f) of 4.85 × 10^8^ Am^−1^ s [100]. On the basis that MF strength could safely be increased given a compensatory decrease in the target region size (i.e., a smaller tumor), Hergt et al., in 2007, proposed the higher threshold of 5.00 × 10^9^ Am^−1^ s^−1^ for a target region with a diameter less than the 30 cm tested by Brezovich [98,99,101]. Up to this point, no clinical studies on MHT in HGGs have surpassed this threshold; the maximum H × f used in two such trials was 1.50 × 10^9^ Am^−1^ s [9,50,75]. Pulsed heating has been shown to minimize the non-specific eddy current heating [14,102,103]. While optimizing AMF parameters is crucial for effective MHT, it is equally important to balance these parameters within safe limits to avoid adverse effects, ensuring that the therapeutic benefits of MHT are maximized without compromising patient safety.

### 2.5. MHT-Mediated Enhancement of Chemotherapy and Radiation

Hyperthermia therapy (HT) has been repeatedly shown to enhance the cytotoxic effects of radiation therapy (RT) on tumor cells. Although the exact mechanism driving this enhancement remains uncertain, it is thought that HT initiates intracellular heat shock responses that disrupt the repair of RT-induced DNA double-strand breaks [104,105,106]. Specifically, it has been found that HT degrades the DNA repair pathway protein BRCA2 [107,108]. Moderate HT has also been shown to enhance perfusion, potentially enhancing the effects of RT by reducing the radioresistant hypoxic cell population. In terms of chemotherapy, hyperthermia has been shown to disrupt the BBB, potentially allowing for increased levels of systemically administered chemotherapeutics to reach the tumor [109,110,111,112]. In addition to enhancing the anticancer effects of chemotherapy and RT through heat, MHT possesses other unique features that further promote tumor sensitization. MIONPs conjugated with GBM-targeting antibodies were shown to radiosensitize and induce apoptosis in the highly therapy resistant stem-like cancer cell populations thought to mediate local GBM recurrence [113]. Additionally, MNPs have been extensively studied as drug carriers that are to deliver chemotherapy directly to the tumor. One group used TMZ-loaded SPIONs to deliver chemotherapy to cancer cells and found that the combination of MHT, chemotherapy, and RT had the greatest anticancer effect compared to any monotherapy or two-modality combination therapy [114]. Many other groups have also shown that MNPs can be effective drug carriers for chemotherapy and found an added anticancer effect when giving MHT in combination with chemotherapy [110,115,116,117,118].

### 2.6. MHT for Glioma Clinical Impact

Over the last three decades, a number of clinical trials have investigated the use of MHT in HGG. A comprehensive list of these is outlined in Table 2. The earliest of these trials was performed in Japan in 1991 by Kobayashi and collaborators [74]. The team conducted a trial implementing MHT on 25 patients with malignant brain tumors, 13 of which were HGGs. MNPs were implanted with either intracavitary implantation or CED, depending on the tumor size. MHT was associated with a positive response rate of 34.8% in HGG patients following treatment, consistent with a complete or partial response rate according to the Japanese Society for Cancer Therapy [119]. The response rate was positive for five of the thirteen patients with HGG (38.4%).

This study was directly followed by two American trials in 1992 and 1994 performed by Stea et al. [86,87]. The 1992 study was a feasibility trial performed in which 28 patients with HGG were treated with a CED infusion of MNPs followed by MHT [87]. The median survival was 20.6 months. The 1994 study compared outcomes in a group of patients treated with MHT plus brachytherapy versus the control group of brachytherapy alone [86]. The 25 patients in the MHT plus brachytherapy group were found to have a 0.53 hazard ratio of death compared to the 37 patients in the control group.

Maier-Hauff et al. published a feasibility study in 2007 [80] followed by an efficacy trial in 2011 [50] where patients with recurrent glioblastoma (GBM) had MNPs stereotactically infused and were then treated with combined MHT and stereotactic RT. The efficacy trial reported a median progression-free survival (PFS) of 13.4 in the 59 GBM (66 total) participants as well as an overall survival (OS) of 23.2 months, both higher than the reported standard-of-care averages (PFS = 6.9 months, OS = 14.6 months, respectively) [50,120]. The brain autopsy studies showed particle aggregation at the sites of installation and the tumor necrosis area. These studies served to catapult MHT into clinical practice in Germany.

Most recently, Grauer et al. [75] implanted MNPs into six recurrent HGG (rHGG) patients through the “NanoPaste” technique, by which the resection cavity wall is coated with layers of Nanotherm^®^ using a hydroxycellulose mesh and fibrin glue. Histopathology specimens of MNP-adjacent tumor areas showed sustained necrosis. The study saw two patients (33%) that experienced a sustained response to treatment, with an overall survival of >23 months.

Overall, the human MHT studies to date have shown consistently positive results. However, there is still much investigation to be carried out to establish MHT as a clinical norm worldwide. In addition to research, increased financial backing and collaboration among academic institutions will be critical for the further development of MHT in the treatment of HGG patients.

## 3. Discussion

MHT provides a potential therapeutic solution for the challenges associated with rHGGs treatment and offers many advantages over other heat-based therapies commonly used to treat brain tumors (i.e., laser interstitial thermal therapy, photothermal therapy). This is in large part due to the fact that, following implantation, MNPs can be remotely activated by an external AMF. The penetration depth of the AMF exceeds that of other activation modalities commonly used in hyperthermia therapy (e.g., light or acoustic waves), allowing for the heating of deeply seated tumors without necessitating further invasive procedures [96]. Moreover, MNPs remain intracranially around the delivery site for weeks to months, potentially allowing for multiple MHT sessions following a single delivery of MNPs [75]. Thus, unlike other thermal therapies, which may only be performed intraoperatively, multiple noninvasive sessions of MHT can be performed after post-operative recovery and the initiation of chemotherapy and RT [121,122,123].

The promising potential of MHT has translated into encouraging preliminary results. Preclinically, MHT has been shown to induce profound antitumor effects and enhance the efficacy of chemotherapy and RT when used to treat HGGs. Furthermore, MNPs have been used as multifunctional theranostic agents in applications such as cancer-targeting drug carriers and MRI contrast agents. Clinically, multiple trials between 1988 and 2019 have shown that numerous sessions of MHT are possible in the brain following a single intracranial delivery of MNPs. Overall, these studies reported that MHT was safe, conferred a survival benefit, and potentially induced an antitumor immune response [9].

Despite these promising initial results and the unique advantages of MHT, clinical progress has slowed over the past decade, particularly in the United States, where the most recent clinical trial took place in 1994 [86]. Additionally, the most well-known European producer of MNPs and AMF generators for clinical use, MagForce, filed for insolvency in 2022. This disparity between the promising preliminary data and the diminishing clinical initiative is puzzling and begs the question of why such a discrepancy exists.

The answer is likely multifactorial. Just as MHT boasts many unique advantages, it faces similarly unique challenges on its path towards clinical application. As described here, the success of MHT depends on the optimization and proper implementation of many distinct and complex components. For instance, designing biocompatible MNPs that are effective for simultaneous MHT and MPI in the brain is a complex process that comprises an entirely separate field of research. The same is true of AMF and MPI device design, as well as the development of intracranial MNP delivery techniques. MHT research spans multiple disciplines and specialties, including physics, bioengineering, cancer biology, clinical medicine, and surgery. As such, researchers with different expertise may work in isolation, leading to slowed progress. In order for any new treatment modality to be effectively translated into a clinical application, interdisciplinary collaboration and cross-talk between these specialists are essential.

This challenge of facilitating such cross-talk is not exclusive to MHT—prior work in other medical specialties has demonstrated the importance of integrating the efforts of researchers and clinicians, who frequently function independently. One example is deep brain stimulation (DBS), a technique in the rapidly evolving subfield of functional neurosurgery that integrates aspects of neurology, psychiatry, neuroscience, and electrical engineering [124,125]. DBS has an established organizational framework for multidisciplinary communication that enables independent regional groups of experts in the aforementioned disciplines to communicate regularly and stay abreast of recent updates [126,127]. Another example is stereotactic radiosurgery (SRS), in which neurosurgeons work in concert with radiation oncologists and medical physicists to plan and provide precise, targeted RT [128,129]. Existing workflows for SRS involve close multidisciplinary collaboration enhanced by the presence of established registries, databases, and conferences that facilitate information sharing [130]. Efforts to advance MHT could benefit greatly from incorporating similar strategies to foster collaboration between the physicists, neurosurgeons, and engineers. The authors suggest organizing an MHT-specific symposium to kick-start crosstalk among neurosurgeons, medical physicists, and other researchers within the field. The goal of this symposium would be to generate connections amongst groups working towards the same goal, clinical adoption.

Beyond addressing this research disconnect, MHT will need to build academic momentum to push into the clinical trial phase. The recent advancement of the chemotherapeutics field may provide valuable precedence. Two technologies discussed above—the CED delivery of MNPs and magnetic MNP targeting—are both the subject of numerous clinical trials studying their compatibility with chemotherapy [131,132,133,134,135]. The favorable clinical trajectory of these shared technologies in the realm of chemotherapeutics can guide the translation of MHT.

Additionally, MHT will need to address shortcomings in federal, philanthropic, and industry funding. Primary brain cancer is rare compared to other types of cancers [136], resulting in relatively less funding. To address this, MHT may benefit by leveraging the steady growth of nanomedicine as a research area and market. At the national level, the National Nanotechnology Initiative (NNI) is an ongoing research and development initiative established by the United States government in 2000, with an initial funding of approximately USD 464 million [137] in 2001. Since then, the nanomedicine subfield has experienced steady growth, with the global nanomedicine sector—valued at USD 53 billion in 2009 [138]—more than tripling by 2022 to USD 170 billion [139]. This major financial investment the field of nanomedicine is projected to receive can potentially address many key obstacles hindering the development of MHT covered here, including the recruitment of experts (and, potentially, the formation of interdisciplinary teams of specialists), the design and production of specialized nanoparticles and AMF generators, and the experimental confirmation of treatment safety and efficacy [140].

Although there are certain challenges facing its clinical application, the future of MHT is bright. The strong body of preliminary pre-clinical and clinical research has repeatedly demonstrated that MHT may confer additional survival benefits, enhance the current standard of care for high-grade brain tumors, and induce cancer cell death through a variety of mechanisms. Significant work is currently being done to address many of the limitations addressed in this review. Previous clinical trials have reported certain therapy-related toxicities due to the amount of delivered magnetic material, the migration of magnetic material during heating, and/or the need to insert invasive temperature probes into the patient’s brain to monitor temperature. At present, numerous groups aim to address these issues by designing MNP constructs optimized for both MPI and MHT, potentially enabling the non-invasive real-time thermometry and imaging of MNPs and more homogenous heating at lower MNP doses. Moreover, human-scale MHT-MPI systems are in development, and trials of these machines are the logical next step in the road towards clinical application. Additional basic science studies are needed to further elucidate the underlying biological mechanisms driving the MHT treatment response and help clinicians to understand how best to integrate MHT into the current standard of care for refractory brain tumors. It is clear that effectively translating MHT from the bench to the clinic is a challenging problem that requires significant progress across multiple disciplines. Although it may appear as if progress has stalled, it is more likely that addressing the shortcomings discovered from previous clinical trials is complex and requires significant research. It is important to recognize that MHT is a highly technology-dependent therapy, and the development of these technologies to the level of clinical use understandably takes time.

## 4. Conclusions

MHT is a promising treatment for HGGs that has not yet realized its full potential despite encouraging preclinical and clinical results. In this review, we described the essential components of the MHT workflow (nanoparticle composition, nanoparticle delivery to areas of interest, and AMF properties and generation), discussing the current state of each, areas of ongoing work, as well as opportunities for future development. We identify multiple factors hindering the clinical translation of MHT, including limited interdisciplinary collaboration and insufficient funding. We propose solutions that draw parallels to other fields in medicine that have experienced rapid evolution in recent years. Efforts to advance MHT in the preclinical and clinical realms are promising and warrant further attention and financial support from academic and industrial stakeholders in order to improve the care and outcomes of patients with HGGs.

## Figures and Tables

**Figure 1 pharmaceuticals-17-00300-f001:**
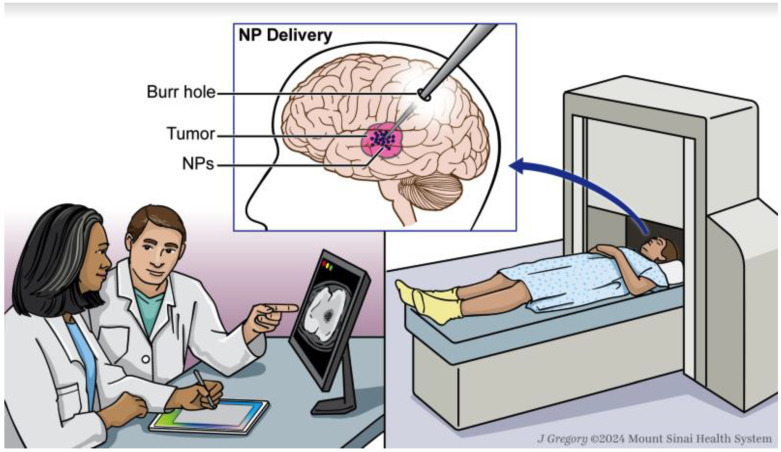
Schematic representation of a proposed MHT clinical workflow: (1) MNP delivery to the targeted lesion, (2) MPI validation of accurate NP delivery, (3) AMF therapy.

**Figure 2 pharmaceuticals-17-00300-f002:**
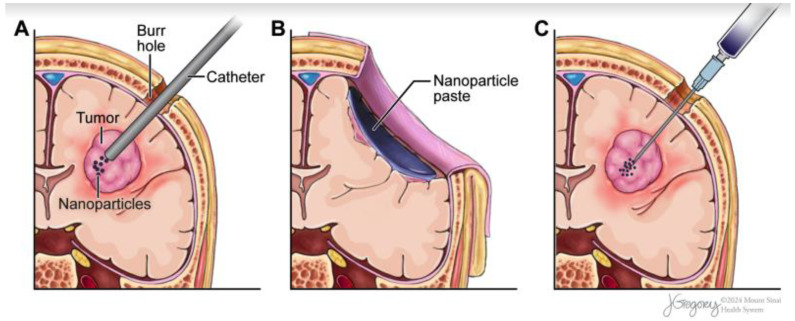
Schematic representation of various MNP delivery methods: (**A**) CED delivery, (**B**) Direct implantation, (**C**) Stereotactic injection.

**Table 1 pharmaceuticals-17-00300-t001:** Magnetic nanoparticles implemented in MHT. Qualitative and quantitative descriptions of their shape, size, composition, and administration. MNPs studied in clinical and preclinical studies with experimental findings.

MNP Type	MNP Shape Illustration	MNP Types Used in Glioma MHT Studies	CharacteristicMNP Size	Study Type	Composition and Administration	ExperimentalResults
Iron Oxide	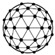 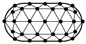 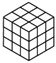 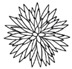	Magnetite (Fe_3_O_4_) [50]	~12 nm (MagForce, used in Maier-Hoff)	Human (phase II clinical trial)	Coated with aminosilane AMF—100 kHz	Demonstrated an overall survival following a diagnosis of 23.2 months in human patients
Iron Oxide	Maghemite (γ-Fe_2_O_3_) [51]	50 nm [52] (Synomag-D)	In vitro and in vivo (mouse model)	AMF—192 kHzIV injection	Delayed tumor growth
Iron Oxide	Hematite, Ferric Oxide (α-Fe_2_O_3_) [53]	3–100 nm	In vivo (mouse model)	PEGylated IV injection	Improved survival
Ferrite	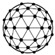	Manganese ferrite [54]	~90 nm	In vivo (mouse model)	IV injection	Treatment effect and low systemic toxicity
Ferrite	Zinc ferrite [55]	~11 nm	In vitro (U-87MG)	AMF—700 kHz	Sustained heating at 41.5 °C to trigger tumor cell death
Ferrite	Mn-Zn ferrite [56]	55 nm	In vitro and in vivo (mouse model)	Rhodamin B isothiocyanate (RBITC)-labeled/mesoporous silica-coatedAMF—160 kHz	Inhibited tumor growth
Other		Manganese oxide (MnO) [57]	120–160 nm	In vitro (U-87MG and U-251 GBM cell lines) and in vivo	IV injection	Demonstrated highly selective cytotoxicity in U-87MG cell lines

**Table 2 pharmaceuticals-17-00300-t002:** Comprehensive summary of human MHT studies for glioma, presented with the treatment population, nanoparticle delivery modality, and study outcomes.

Study Authors	Title	N (#)	MNP Delivery Modality	MNPs Used	Study Outcomes
Kobayashi et al., 1991 [74]	Interstitial hyperthermia of malignant brain tumors by implant heating system: clinical experience	25	Direct Implantation + CED	Fe-Pt Alloy	Successful treatment completion in 23 of 25 patients with a 34.8% overall response rate to treatment
Stea et al., 1992 [87]	Treatment of malignant gliomas with interstitial irradiation and hyperthermia	28	CED	Ni-S Alloy	Demonstrated feasibility of the interstitial MHT of brain tumors with ferromagnetic implants, with a median patient survival of 20.6 months from diagnosis
Stea et al., 1994 [86]	Interstitial irradiation versus interstitial thermoradiotherapy for supratentorial malignant gliomas: a comparative survival analysis	62	CED	Ni-S Alloy	The hazard of dying when treated with hyperthermia plus brachytherapy was 0.53 times that of the control group treated with brachytherapy alone
Maier-Hauff et al., 2007 [80]	Intracranial Thermotherapy using Magnetic Nanoparticles Combined with External Beam Radiotherapy: Results of a Feasibility Study on Patients with Glioblastoma Multiforme	14	Stereotactic Injection	Aminosilane-coated Fe_3_O_4_	Treatment with a median maximum intratumoral temperature of 44.6 degrees C was tolerated in all 14 patients
Maier-Hauff et al., 2011 [50]	Efficacy and safety of intratumoral thermotherapy using magnetic iron-oxide nanoparticles combined with external beam radiotherapy on patients with recurrent glioblastoma multiforme	66	Stereotactic Injection	Aminosilane coated Fe_3_O_4_	An overall survival after a primary tumor diagnosis of 23.4 months and an overall survival following a diagnosis of first tumor recurrence of 13.4 months
Grauer et al., 2019 [75]	Combined intracavitary thermotherapy with iron oxide nanoparticles and radiotherapy as local treatment modality in recurrent glioblastoma patients	6	Direct Implantation	Aminosilane-coated Fe_3_O_4_	Demonstrated inflammatory reaction surrounding the resection cavity following intracavitary MHT in combination with radiation therapy, potentially triggering a potent antitumor immune response

## Data Availability

No new data were created for this research.

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
