# Peer review of "Magnetic Hyperthermia Therapy for High-Grade Glioma: A State-of-the-Art Review"

_pharmaceuticals, 2024, doi:10.3390/ph17030300_

Round 1

Reviewer 1 Report

Comments and Suggestions for Authors

Review Report

Ms.: pharmaceuticals-2865009 

This review aims to present the relevant results from "Magnetic Hyperthermia Therapy for High Grade Glioma", as the title suggests. The field of alternative cancer therapy through the MHT method, and in particular of glioma therapy, is of high interest, this being a very promising method in this field due to its effectiveness and reduced toxicity compared to traditional methods (chemo- and radiotherapy) which have a high degree of toxicity on the body. From this point of view, this review is welcome to update the results in this field.

However, before it is published, a major revision is needed to complete this review, so that it reproduces as well and completely as possible the results obtained so far in this MHT therapy on glioma. Below are the comments for the manuscript revision:

Major remarks:

1. In the "Nanoparticles" section, a table must be inserted with the representative, quantitative, and synthetic data, regarding the magnetic nanoparticles used in MHT: their characteristic sizes, biocompatibility, and results obtained in vitro/vivo regarding the toxicity and effect of MHT on tumors;

2. Also in this section, it should be clearly stated which magnetic nanoparticles (type, size, shapes, biocompatibility, etc.) are currently used in MHT of glioma, and which are the best results obtained with them in vitro in vivo and clinical trials (if any);

3. Having in view that the review refers to MHT therapy, it is necessary that the authors complete at the end (after the AMF section) a new chapter referring to this issue, and the results obtained so far in the treatment of glioma (in vitro/in vivo);

4. To support the results presented in the entire review, the authors need to introduce more relevant figures in their manuscript, especially in the "Nanoparticle Delivery" sections and the new section (MHT in glioma).

Minor remark:

5. Sections must be numbered, in accordance with Pharmaceuticals journal rules.

Author Response

Dear Reviewer,

The co-authors would like to express foremost gratitude to you for the time you spent reading our manuscript and providing feedback. We have listed below each of your comments along with a response from the team as to how the comment was addressed in the most updated manuscript. We trust that you will find our updates acceptable and we are very proud of the work we are presenting. Thank you once again for your time.

  1. In the "Nanoparticles" section, a table must be inserted with the representative, quantitative, and synthetic data, regarding the magnetic nanoparticles used in MHT: their characteristic sizes, biocompatibility, and results obtained in vitro/vivo regarding the toxicity and effect of MHT on tumors.

A table (Table 1) listing the nanoparticles’ size, biocompatibility, toxicity, results, and shapes has been added

  1. Also in this section, it should be clearly stated which magnetic nanoparticles (type, size, shapes, biocompatibility, etc.) are currently used in MHT of glioma, and which are the best results obtained with them in vitro in vivo and clinical trials (if any).

The authors have made a specific note of which NPs were used in glioma trials, as well as adding the NP used in each human trial to Table 2

  1. Having in view that the review refers to MHT therapy, it is necessary that the authors complete at the end (after the AMF section) a new chapter referring to this issue, and the results obtained so far in the treatment of glioma (in vitro/in vivo).

The authors have added a section directly after AMF to summarize pre-clinical and clinical results for the MHT intervention as a whole.

  1. To support the results presented in the entire review, the authors need to introduce more relevant figures in their manuscript, especially in the "Nanoparticle Delivery" sections and the new section (MHT in glioma).

Table 2 has been updated to coincide with the use of MHT in glioma treatment, and a figure illustrating the MNP delivery techniques has been added as well.

Minor remark:

  1. Sections must be numbered, in accordance with Pharmaceuticals journal rules.

The authors added section numbers

Thank you,

The co-authors

Reviewer 2 Report

Comments and Suggestions for Authors

This paper is a brief overview on some major aspects of the therapy of high grade glioma by magnetic hyperthermia. This is a very useful paper, and it is recommended to be accepted for publication after some revision on the basis of comments below.

The information on the different aspects of the magnetic hyperthermia therapy (MHT) is described by the authors in a quite compressed way, and therefore some parts are hard to follow. Therefore, it would be beneficial to summarize all the major results, achievements and current state of all segments of this field in tables, similar to Table 1. These should include the nanoparticles and their assemblies, coatings to provide biocompatibility and avoiding aggregation, their hyperthermia and MPI applications, their delivery methods, the AMF details, and obviously the major results and open problems related to these aspects of MHT and related theranostics as well.

For such issues, the authors are recommended to see and cite the following publications and the related references therein:

Zhang, L.; Li, Q.; Liu, J.; Deng, Z.; Zhang, X.; Alifu, N.; Zhang, X.; Yu, Z.; Liu, Y.; Lan, Z.; Wen, T.; Sun, K. (2024). Recent advances in functionalized ferrite nanoparticles: from fundamentals to magnetic hyperthermia cancer therapy. Coll. Surf. B: Biointerfaces 2024, 234, 113754.

Wu, H.; Liu, L.; Ma, M.; Zhang, Y. Modulation of blood-brain tumor barrier for delivery of magnetic hyperthermia to brain cancer. J. Contr. Rel. 2023, 355, 248-258.

Zachou, M.-E.; Kouloulias, V.; Chalkia, M.; Efstathopoulos, E.; Platoni, K. The Impact of Nanomedicine on Soft Tissue Sarcoma Treated by Radiotherapy and/or Hyperthermia: A Review. Cancers 202416, 393.

Farzanegan, Z.; Tahmasbi, M. Evaluating the applications and effectiveness of magnetic nanoparticle-based hyperthermia for cancer treatment: A systematic review. Appl. Rad. Isotopes, 2023, 198, 110873.

Dhar, D.; Ghosh, S.; Das, S.; Chatterjee, J. A review of recent advances in magnetic nanoparticle-based theranostics of glioblastoma. Nanomedicine 2022, 17, 107-132.

Wlodarczyk, A.; Gorgon, S.; Radon, A.; Bajdak-Rusinek, K. Magnetite nanoparticles in magnetic hyperthermia and cancer therapies: Challenges and perspectives. Nanomaterials 2022, 12, 1807.

Hou, Z.; Liu, Y.; Xu, J.; Zhu, J. Surface engineering of magnetic iron oxide nanoparticles by polymer grafting: synthesis progress and biomedical applications. Nanoscale 202012, 14957-14975.

Illes, E.; Szekeres, M.; Toth, I. Y.; Szabo, A; Ivan, B.; Turcu, R.; Vekas, L.; Zupko, I.; Jaics, G.; Tombacz, E. Multifunctional PEG-carboxylate copolymer coated superparamagnetic iron oxide nanoparticles for biomedical application. J. Magn. Magn. Mater. 2018, 451, 710-720.

Israel, L. L.; Galstyan, A.; Holler, E.; Ljubimova, J. Y. Magnetic iron oxide nanoparticles for imaging, targeting and treatment of primary and metastatic tumors of the brain. J. Contr. Rel. 2020, 320, 45-62.

Montiel Schneider, M. G.; Martín, M. J.; Otarola, J.; Vakarelska, E.; Simeonov, V.; Lassalle, V.; Nedyalkova, M. Biomedical applications of iron oxide nanoparticles: Current insights progress and perspectives. Pharmaceutics, 2022, 14, 204.

Peréz, D. L.; Puentes, I.; Romero, G. A.; Gaona, I. M.; Vargas, C. A.; Rincón, R. J. Synthesis of superparamagnetic iron oxide nanoparticles coated with polyethylene glycol as potential drug carriers for cancer treatment. J. Nanoparticle Res. 2024, 26, 1-15.

Dulinska-Litewka, J.; Lazarczyk, A.; Halubiec, P.; Szafranski, O.; Karnas, K.; Karewicz, A.  Superparamagnetic iron oxide nanoparticles – Current and prospective medical applications. Materials2019, 12, 617.

Feng, Q.; Liu, Y.; Huang, J.; Chen, K.; Huang, J.; Xiao, K. Uptake, distribution, clearance, and toxicity of iron oxide nanoparticles with different sizes and coatings. Sci. Reports2018, 8, 1-13.

Martinkova, P.; Brtnicky, M.; Kynicky, J.; Pohanka, M. Iron oxide nanoparticles: innovative tool in cancer diagnosis and therapy. Adv. Healthcare Mater. 2018, 7, 1700932.

As for the desired progress in the field of MHT, similar to the cooperation in the area of deep brain stimulation (DBS) mentioned on page 6, it is proposed that the authors draw a scheme for the collaboration of institutions and people at interconnecting scientific, technical and medical fields, which might get broad attention for the readers of this review.

Author Response

Dear Reviewer,

The co-authors would like to express foremost gratitude to you for the time you spent reading our manuscript and providing feedback. We have listed below each of your comments along with a response from the team as to how the comment was addressed in the most updated manuscript. We trust that you will find our updates acceptable and we are very proud of the work we are presenting. Thank you once again for your time.

The information on the different aspects of the magnetic hyperthermia therapy (MHT) is described by the authors in a quite compressed way, and therefore some parts are hard to follow. Therefore, it would be beneficial to summarize all the major results, achievements and current state of all segments of this field in tables, similar to Table 1.

The authors have added a section titled “MHT for Glioma Clinical Impact” that summarizes results in clinical and preclinical trials. Table 2 has been updated accordingly as well. 

These should include the nanoparticles and their assemblies, coatings to provide biocompatibility and avoiding aggregation, their hyperthermia and MPI applications, their delivery methods, the AMF details, and obviously the major results and open problems related to these aspects of MHT and related theranostics as well.

Table 1 has been created to display NP types as well as size, biocompatibility, toxicity, results, and shapes. Table 2 has been updated to list delivery type used and NP used for all human trials.

As for the desired progress in the field of MHT, similar to the cooperation in the area of deep brain stimulation (DBS) mentioned on page 6, it is proposed that the authors draw a scheme for the collaboration of institutions and people at interconnecting scientific, technical and medical fields, which might get broad attention for the readers of this review.

The authors have added a proposition for an MHT symposium to address the lack of crosstalk between researchers in this field.

Thank you, 

The co-authors

Reviewer 3 Report

Comments and Suggestions for Authors

The authors present a review which focus on the use of Magnetic Hyperthermia as a means to treat High grade Glioma both as a main and adjuvant therapy. I would say that the review is sound, and that the discussion section, in particular, raises quite relevant points to the scientific community. Still, this review falls behind in some areas and approaches that must be addressed before being considered to publication.

1)      In the abstract the authors state: “MHT may enhance the effectiveness of chemotherapy and radiation therapy (RT) for the treatment of brain tumors” still there is a clear lack of a dedicated section/paragraph where the authors expand the synergy of MHT with radiotherapy. In fact, there are plenty of works that address this branch of Hyperthermia, namely:

[A] Spirou, Spiridon V., et al. "Recommendations for in vitro and in vivo testing of magnetic nanoparticle hyperthermia combined with radiation therapy." Nanomaterials 8.5 (2018): 306.

[B] Peeken, Jan C., Peter Vaupel, and Stephanie E. Combs. "Integrating hyperthermia into modern radiation oncology: what evidence is necessary?." Frontiers in oncology 7 (2017): 132.

[C] Horta, André C., et al. "High yttrium retention in magnetite nanoparticles functionalized with hybrid silica-dextran shells." Nano-Structures & Nano-Objects 36 (2023): 101065.

[D] Datta, Niloy R., et al. "Hyperthermia and radiotherapy in the management of head and neck cancers: A systematic review and meta-analysis." International Journal of Hyperthermia 32.1 (2016): 31-40.

[E] Ognjanović, Miloš, et al. "99mTc-, 90Y-, and 177Lu-labeled iron oxide nanoflowers designed for potential use in dual magnetic hyperthermia/radionuclide cancer therapy and diagnosis." ACS applied materials & interfaces 11.44 (2019): 41109-41117.

2)      Page 2 authors wrote “adjunct”. It should be adjuvant.

3)      In the Nanoparticles section the authors talk about the Specific Loss Power but forgot to mention the Intrinsic Loss Power (ILP), which is a more recent but more adequate way of describing the potential for MHT.

4)      In the same section the authors mention that it is common to coat the nanoparticles with silica, but this approach would reduce the saturation magnetization by 32%. In the work of Fernandes et al. [F], it is shown that this effect can be as low as a reduction of less than 8% in the saturation magnetization of the sample (total mass) and that in fact the saturation magnetization of the core (normalized by the mass of magnetite) can actually increase as much as 14%.

[F] Fernandes, Tiago, et al. "Chemical Strategies for Dendritic Magneto‐plasmonic Nanostructures Applied to Surface‐Enhanced Raman Spectroscopy." Chemistry–A European Journal 28.61 (2022): e202202382.

5)      In page 3, in the MPI section the description of this technique/approach is too brief. The MPI should be explained in more detail. For example, the authors stated that “The underlying physics of MPI can further be applied to enable real-time, noninvasive magnetic nanothermometry (MNT) during MHT.” But do not expand how.

6)      Still regarding the last point, there is also an additional problem in the MPI section which is also transversal to the remaining of the manuscript. There is a clear lack of figures/schemes that could better clarify the points and explanations raised by the authors. Besides illustrative figures for the MPI section, figures that help the explanation of the Nanoparticles, Nanoparticle Delivery and AMF sections should also be added.

7)      Some acronyms are not defined in the text. For example, BBB (blood-brain barrier) is not defined in the text (page 3, in the beginning of the Nanoparticle Delivery section). The acronyms LITT and PDT, found in the end of page 4 should also be defined.

8)      In the end of page 4 the authors state “The efficacy trial reported a median progression free survival (PFS) of 13.4 in the 59 GBM (66 total) participants as well as an overall survival (OS) of 23.2 months, both higher than reported standard of care averages.” The authors should specify with are the reported standard of care averages.

9)      The AMF section should be improved in order to include a better description of the relaxation mechanisms (describing Brown and Neel relation mechanisms, as well as the Rosensweig model to determine the SLP/ILP, using figures and equations). In this section there should also be mention the relevance of the superparamagnetic regime in these mechanisms.

10) In the same section there is a language abuse while describing the heating mechanisms, namely by dividing the heat generation into hysteresis loss and thermal relaxation sources. Technically all heating generation arises from hysteresis losses. In the case of the superparamagnetic particles, while it seems to be the case of a null hypothesis, in fact there is a phase difference between the magnetization and the applied field which results in a hysteretic behavior, hence there is also a dynamic hysteresis loss. Furthermore, when the authors describe the hysteresis loss mechanism, they do it poorly. The friction that they mention somehow can induce the reader to interpret it as a mechanical friction which is not the case. The hysteretic losses that arise from the hysteresis measured in the “static” curves come from the creation and destruction of magnetic domains, and not some kind of macroscopic friction as referred by the authors (this effect is NOT macroscopic, it clearly has microscopic origins). Hence this discussion/description must be improved.

Author Response

Dear Reviewer,

The co-authors would like to express foremost gratitude to you for the time you spent reading our manuscript and providing feedback. We have listed below each of your comments along with a response from the team as to how the comment was addressed in the most updated manuscript. We trust that you will find our updates acceptable and we are very proud of the work we are presenting. Thank you once again for your time.

  1. In the abstract the authors state: “MHT may enhance the effectiveness of chemotherapy and radiation therapy (RT) for the treatment of brain tumors” still there is a clear lack of a dedicated section/paragraph where the authors expand the synergy of MHT with radiotherapy.

A dedicated paragraph has been added to describe the mechanisms of hyperthermia-mediated enhancement of radiotherapy and chemotherapy, as well as the unique ways in which MHT enhances these two therapies.

  1. Page 2 authors wrote “adjunct”. It should be adjuvant.

The authors have corrected this error.

  1. In the Nanoparticles section the authors talk about the Specific Loss Power but forgot to mention the Intrinsic Loss Power (ILP), which is a more recent but more adequate way of describing the potential for MHT.

The authors have added an explanation of ILP, as well as why it is superior to SLP. 

  1. In the same section the authors mention that it is common to coat the nanoparticles with silica, but this approach would reduce the saturation magnetization by 32%. In the work of Fernandes et al. [F], it is shown that this effect can be as low as a reduction of less than 8% in the saturation magnetization of the sample (total mass) and that in fact the saturation magnetization of the core (normalized by the mass of magnetite) can actually increase as much as 14%. 

A sentence has been added to this section to account for the work of Fernandes et al., which provides evidence for potentially increased, as opposed to decreased, saturation magnetization after silica coating of MNPs.

  1. In page 3, in the MPI section the description of this technique/approach is too brief. The MPI should be explained in more detail. For example, the authors stated that “The underlying physics of MPI can further be applied to enable real-time, noninvasive magnetic nanothermometry (MNT) during MHT.” But do not expand how.

This section has been expanded to further elaborate on the technique and principles of MPI, its advantages, as well as the ways in which the properties of MPI can be applied for nanothermometry.

  1. Still regarding the last point, there is also an additional problem in the MPI section which is also transversal to the remainder of the manuscript. There is a clear lack of figures/schemes that could better clarify the points and explanations raised by the authors. Besides illustrative figures for the MPI section, figures that help the explanation of the Nanoparticles, Nanoparticle Delivery and AMF sections should also be added. 

The authors have added figures to better display concepts covered in the review. 

  1. Some acronyms are not defined in the text. For example, BBB (blood-brain barrier) is not defined in the text (page 3, in the beginning of the Nanoparticle Delivery section). The acronyms LITT and PDT, found at the end of page 4 should also be defined.

The authors have defined all of the acronyms included in the text.

  1. In the end of page 4 the authors state “The efficacy trial reported a median progression free survival (PFS) of 13.4 in the 59 GBM (66 total) participants as well as an overall survival (OS) of 23.2 months, both higher than reported standard of care averages.” The authors should specify which are the reported standard of care averages.

The authors have added the SOC averages as reported in the NEJM.

  1. The AMF section should be improved in order to include a better description of the relaxation mechanisms (describing Brown and Neel relaxation mechanisms, as well as the Rosensweig model to determine the SLP/ILP, using figures and equations). In this section there should also be mention of the relevance of the superparamagnetic regime in these mechanisms. 

A better description of relaxation mechanisms has been added by the authors. Including an explanation of ILP and SLP as well as Rosenwieg’s model.

  1. In the same section there is a language abuse while describing the heating mechanisms, namely by dividing the heat generation into hysteresis loss and thermal relaxation sources. Technically all heating generation arises from hysteresis losses. In the case of the superparamagnetic particles, while it seems to be the case of a null hypothesis, in fact there is a phase difference between the magnetization and the applied field which results in a hysteretic behavior, hence there is also a dynamic hysteresis loss. Furthermore, when the authors describe the hysteresis loss mechanism, they do it poorly. The friction that they mention somehow can induce the reader to interpret it as a mechanical friction which is not the case. The hysteretic losses that arise from the hysteresis measured in the “static” curves come from the creation and destruction of magnetic domains, and not some kind of macroscopic friction as referred to by the authors (this effect is NOT macroscopic, it clearly has microscopic origins). Hence this discussion/description must be improved.

The authors have refined the language used to describe these phenomena, and clarified that heat generation arises from hysteresis loss. The concept of friction has been more specifically linked to Brownian motion.

Thank you,

The co-authors

Reviewer 4 Report

Comments and Suggestions for Authors

This manuscript concerns with the description of magnetic hyperthermia therapy (MHT) for the treatment of high-grade glioma (HGG) in general terms. Issues related to optimizing the shape, size and coating of magnetic nanoparticles (MNPs) are briefly reflected, injection methods of MNPs into the brain are described (direct intracavitary implantation after the surgical resection, as well as convection enhanced delivery (CED) using a stereotactically placed catheter, or direct stereotactic injection), general parameters under which MHT is performed, as well as problems of post-procedural imaging of nanoparticles (MPI as an alternative to MRI). The authors raised questions about the problems of MHT, including the current low efficiency of this method and problems with its further development. The review is logical and written in good language. However, after reading this review, there is the impression that the manuscript is poorly filled with factual and illustrative material. Perhaps the authors should expand on the topics raised in the manuscript in more depth, using more recent references (after 2019, which are currently ~30% references in the manuscript), and also provide more figures and schemes to make the material more easy readable and attractive to expand the readership.

The manuscript can be published in Pharmaceuticals, but after a major revision.

Author Response

Dear Reviewer,

The co-authors would like to express foremost gratitude to you for the time you spent reading our manuscript and providing feedback. We have listed below each of your comments along with a response from the team as to how the comment was addressed in the most updated manuscript. We trust that you will find our updates acceptable and we are very proud of the work we are presenting. Thank you once again for your time.

The authors have addressed the above reviewers’ comments, as well as added figures to better display concepts covered in the review.

Thank you,

The co-authors

Round 2

Reviewer 1 Report

Comments and Suggestions for Authors

The authors responded to all my comments, completing and improving their manuscript accordingly, and now the revised manuscript can be published in the journal.

Author Response

Dear reviewer, thank you for your feedback and acceptance.

Sincerely,

The co-authors

Reviewer 3 Report

Comments and Suggestions for Authors

The authors have addressed most of my comments. I think that the current version of the manuscript is acceptable for publication.

Author Response

(The authors gave the same response as above.)

Reviewer 4 Report

Comments and Suggestions for Authors

The authors have significantly improved their work. However, to refresh it, I still recommend increasing the number of references after 2019, which in this version of the manuscript constitute only ~30% of the total number of references.

The manuscript may be published in Pharmaceuticals.

Author Response

Dear Reviewer, 

The authors have updated some of the citations to reflect a more current references list. 

Thank you for your feedback,

The co-authors